# Impact on Glucose Homeostasis: Is Food Biofortified with Molybdenum a Workable Solution? A Two-Arm Study

**DOI:** 10.3390/nu14071351

**Published:** 2022-03-24

**Authors:** Sonya Vasto, Francesca Di Gaudio, Maria Raso, Leo Sabatino, Rosalia Caldarella, Claudio De Pasquale, Luigi Di Rosa, Sara Baldassano

**Affiliations:** 1Department of Biological, Chemical and Pharmaceutical Sciences and Technologies (STEBICEF), University of Palermo, Viale delle Scienze, 90128 Palermo, Italy; luigi.dirosa@unipa.it; 2Euro-Mediterranean Institutes of Science and Technology (IEMEST), 90139 Palermo, Italy; 3Department of Promoting Health, Maternal-Infant, Excellence and Internal and Specialized Medicine (ProMISE) G. D’Alessandro, University of Palermo, 90127 Palermo, Italy; francesca.digaudio@unipa.it; 4Chromatography and Mass Spectrometry Section, Quality Control and Chemical Risk (CQRC), Department PROMISE, University Palermo, 90133 Palermo, Italy; rasomaria.mr@libero.it; 5Dipartimento Scienze Agrarie, Alimentari e Forestali (SAAF), University of Palermo, Viale delle Scienze, Ed. 5, 90128 Palermo, Italy; leo.sabatino@unipa.it (L.S.); claudio.depasquale@unipa.it (C.D.P.); 6Department of Laboratory Medicine, “P. Giaccone” University Hospital, 90128 Palermo, Italy; liacaldarella@virgilio.it

**Keywords:** biofortification, lettuce, human heath, minerals, gut peptides, GIP, PYY

## Abstract

Diabetes is expected to increase up to 700 million people worldwide with type 2 diabetes being the most frequent. The use of nutritional interventions is one of the most natural approaches for managing the disease. Minerals are of paramount importance in order to preserve and obtain good health and among them molybdenum is an essential component. There are no studies about the consumption of biofortified food with molybdenum on glucose homeostasis but recent studies in humans suggest that molybdenum could exert hypoglycemic effects. The present study aims to assess if consumption of lettuce biofortified with molybdenum influences glucose homeostasis and whether the effects would be due to changes in gastrointestinal hormone levels and specifically Peptide YY (PYY), Glucagon-Like Peptide 1 (GLP-1), Glucagon-Like Peptide 2 (GLP-2), and Gastric Inhibitory Polypeptide (GIP). A cohort of 24 people was supplemented with biofortified lettuce for 12 days. Blood and urine samples were obtained at baseline (T0) and after 12 days (T2) of supplementation. Blood was analyzed for glucose, insulin, insulin resistance, β-cell function, and insulin sensitivity, PYY, GLP-1, GLP-2 and GIP. Urine samples were tested for molybdenum concentration. The results showed that consumption of lettuce biofortified with molybdenum for 12 days did not affect beta cell function but significantly reduced fasting glucose, insulin, insulin resistance and increased insulin sensitivity in healthy people. Consumption of biofortified lettuce did not show any modification in urine concentration of molybdenum among the groups. These data suggest that consumption of lettuce biofortified with molybdenum improves glucose homeostasis and PYY and GIP are involved in the action mechanism.

## 1. Introduction

Type 2 diabetes—(T2D) is a disease strongly associated with lifestyle [1]. According to the 2019 International Diabetes Federation (IDF) which promotes diabetes care, prevention and a cure, by 2045, it is expected that this disease would increase to 700 million people worldwide. The use of nutritional interventions is a natural approach for managing diabetes [2] that could help in reducing drug prescriptions and hospitalizations.

Minerals are of paramount importance in order to preserve and obtain good health and their key role is supported by a significant amount of literature [3]. The crucial role of minerals shifts from helping vitamin absorption [4] and immunomodulatory effect [5] to metabolic [6] and bone metabolism [7], influencing hematological [8] and endocrine function [9]. The source of many important minerals is mainly plant-based therefore it is important to use a balanced diet to provide the adequate proportion of minerals [10]. 

Molybdenum is an essential component of four enzymes in mammals: xanthine oxidoreductase or xanthine oxidase/dehydrogenase, aldehyde oxidase, sulfite oxidase and mARC (mitochondrial amidoxyme reducing component) [11]. The mammals molybdenum enzymes of mARC [12] are the most recently discovered in mammalian and belong to the sulfite family of molybdenum-containing enzymes able to catalyze a large range of redox-reactions. Xanthine dehydrogenase, aldehyde oxidase, sulfite oxidase and mARC harbor a pterin-based molybdenum cofactor (Moco) in their active site [13]. These four enzymes are involved in the metabolism of aromatic aldehydes, in the catabolism of sulfur amino acids and heterocyclic compounds such as purines, pyrimidines, pteridines and pyridines. There are different sources of molybdenum as in legumes, whole grains, leafy vegetables, milk and cheese. Molybdenum deficiency is rare, since this element is taken in sufficient quantities through food [10]. Only one clinical case of molybdenum deficiency is known, in literature, due to the absence of molybdenum on long term total parenteral nutrition in patients suffering of Crohn’s disease [14] while a deficiency of molybdenum cofactor dependent enzymes is due to an autosomal recessive disease called molybdenum cofactor deficiency [15]. The US Institute of Medicine has estimated an average requirement (EAR) of 22 μg/day of molybdenum with the addition of 3 μg/day to compensate for daily losses. However, if we consider the bioavailability of molybdenum that could vary from one study to another, the EAR goes to 34 μg/day, assuming that the mean bioavailability of molybdenum is 75%. So, at the end the recommended dietary allowance (RDA) is 45 μg/day. The consumption of a diet high in molybdenum usually does not pose a health risk because the molybdenum is rapidly excreted in urine [11]. The tolerable upper intake level (UL) for molybdenum is 2 mg/day. The U.S. National Health and Nutrition Examination Survey (NHANES) and the Canadian Health Measures Survey (CHMS) established the biomonitoring equivalents (BEs) values that are estimates of the concentration of a chemical or its metabolite in blood or urine that are consistent with defined exposure guidance values such reference dose (RfD), and tolerable daily intake (TDI) for the population. The BE molybdenum values in plasma and urine are 0.5, and 22 μg/L, respectively, while the BE values associated with toxicity that in plasma range from 0.9 to 31 μg/L and in urine 200–7500 μg/L [16]. 

Recent studies in humans suggest that molybdenum could exert hypoglycemic effects [17,18]. However, although there are reports on leafy green vegetables biofortification with trace elements [19,20,21,22] including molybdenum [23,24,25], there are no studies about the consumption of biofortified food with molybdenum on glucose homeostasis in individuals and the mechanism of action has never been investigated. In this manuscript it was hypothesized that the consumption of biofortified food with molybdenum affects glucose homeostasis by influencing endogenous gut hormones in the healthy population. Therefore, the present study aims to assess if the consumption of lettuce biofortified with molybdenum influences glucose homeostasis and whether the effects would be due to changes in endogenous gastrointestinal hormones and specifically Peptide YY (PYY), Glucagon-Like Peptide 1 (GLP-1), Glucagon-Like Peptide 2 (GLP-2), and Gastric Inhibitory Polypeptide (GIP). 

## 2. Materials and Methods

### 2.1. Trial Setup, Plant Materials, Nutraceutical Traits and Crop Management

In this study molybdenum -enrichment was made by supplying molybdenum in the form of sodium molybdate (Na_2_MoO_4_) provided through foliar spray during the period of growth, as described by Sabatino et al. [24]. After the harvesting season, the endive crops were provided for human consumption in a selected population. In details: 100 g of canasta lettuce was consumed by a cohort of 24 individuals and blood and urine samples were obtained at the beginning (T0) and after 12 days of consumption (T2) (Figure 1). With regard to molybdenum determination, the molybdenum content in leaves and urine samples were assessed via inductively coupled plasma mass spectrometry (ICP-MS). The lettuce employed in the current study was cultivated following the work of Sabatino and collaborators [19], excluding the protein hydrolysate treatment. 

### 2.2. Study Design

A cohort of 24 healthy subjects (12 men and 12 women) were recruited at Policlinico Hospital “Paolo Giaccone Palermo”. The volunteers were Sicilians, living in the area of West-Sicily. A group of well-trained nutritionists and physicians administered a questionnaire to collect anamnestic data of interest, food habits (24 h recall) and lifestyle. Participants were selected following inclusion and exclusion criteria (Table 1). The volunteers were advised to not use any product containing molybdenum, dietary supplements. We advised the cohort to follow the same regular nutritional pattern.

An informed consent was signed by participants before enrolment and they were identified with an alphanumeric number to respect privacy. The study protocol was conducted in accordance with the Declaration of Helsinki. It was approved by the ethic committee of Palermo university hospital (biofortification, No.2/2020) AIFA CE 150109 and the trial is registered at Clinicaltrials.gov NCT04985240 (Nutri-Mo-Food). The canasta lettuce plants (in total at least 2 kg) were given to each participant in order to utilize 100 g every day for a period of 12 days. The treatment duration was chosen on the basis of our previous study that reported no negative effect after 12 days of nutritional intervention with iodine biofortified curly endive [9]. Plants were stored at 4 ± 1 °C. Healthy volunteers were randomly assigned to the groups; 12 (6 males and 6 females) received control canasta lettuce (control group) and 12 (6 males and 6 females) received canasta lettuce with molybdenum (biofortified group). Both cohorts were asked to write a food diary during the study.

At baseline (T0 = BASE) and after 12 days (T2) blood and urine samples were obtained. The healthy volunteers were advised to not use any food supplementation or integration 20 days before the baseline and for the entire period of 12 days of canasta lettuce administration. The healthy volunteers attended the first visit and underwent anthropometric measurement and 24-h dietary recall. Daily nutritional intake was recorded over a period of 8 consecutive days before starting the study and until the end of the experiment (food diary). This first period was assumed to be representative of the healthy volunteer’s habitual nutritional intake. The healthy volunteers were provided with a food diary and instructed to record all food and beverages (including quantities) consumed over the 8 days before starting the study and until the end of the study. 

Participants underwent vein puncture in the morning (between 7.00 and 8.00 a.m.) in fasting state and urine samples were collected [9]. Hematochemical tests were performed at the central laboratory analysis of Palermo university hospital according to standard procedure at baseline (BASE) and after 12 days (Figure 1). Body weight, barefoot standing height, body mass index, body composition was measured in the different study groups [26]. 

### 2.3. Molybdenum Determination 

To examine molybdenum content, air-dried lettuce samples were ground in a variable speed rotor mill Pulverisette (Idar-Oberstein, Alemania, Germany). Samples of lettuce (0.5 g) were digested in the mixture of 10 cm^3^ 65% HNO_3_ and 0.8 cm^3^ 30% H_2_O_2_ were conducted in the microwave system MARS-5 Xpress (CEM, World Headquarters, Matthews, NC, USA). The content of molybdenum in urine was examined after 10 times sample dilution with ultrapure water (milliQ) without any digestion step by Inductively Coupled Plasma Mass Spectrometry (ICP-MS) (X Series II, Thermo Fisher Scientific, Rodano, Italy).

### 2.4. Calibration

Calibration standards for molybdenum were arranged on a daily basis by stepwise dilution of the molybdenum standard 1000 mg/L in a 1% HNO_3_ medium to yield final concentrations of 0.05, 0.1, 0.5, 0.75, 1.0, 5.0, 10.0, 25.0, 50.0, 100.0, 250.0, 500 mg/L. The molybdenum ion was determined at *m*/*z* 95. The detection limit was assessed as 3 standard deviation (SD) of the concentration in the urine sample. Quantitative determinations for molybdenum were assessed by standard addition method on a daily basis by stepwise dilution of the MoNa solution in a 10 times diluted urine to yield final concentration of 20, 30, 40, 50, 75, 100, 250 µg/L. Solution containing, Y (50 µg L^−1^) was used as internal standards to compensate for any signal instability or sensitivity changes during the analysis. A solution of HNO_3_ 2% as blank was used. The molybdenum ion was determined at *m*/*z* 127. The detection limit was determined as 3 SD of the concentration in the urine sample. 

### 2.5. Biochemical Analysis

To quantify gastrointestinal hormones the plasma samples were collected as previously shown [27,28] and were analyzed in duplicates using the enzyme immunoassay kits for active GLP-1 (EZGLPHS-35K), total GIP (EZHGIP-54K), total PYY (EZGRT-89K), GLP-2 (EZGLP-237K) from Millipore [29]. Fasting glucose and insulin, insulin resistance (HOMA2-IR), β-cell function (HOMA2-%B) and insulin sensitivity (HOMA2-%S) were measured as previously shown [30]. 

### 2.6. Statistical Analyses 

For power calculation, the inclusion of eight subjects would be needed to be able to detect differences in fasting glucose (probability [β] of 20% and level of statistical significance [α] of 5%) based on our previous study [30]. In this case, 12 subjects were included in order to decrease the risks of type 2 errors and to increase the power for evaluation of secondary outcomes. Student t tests were used to compare the baseline characteristics of the two groups. Changes between baseline and follow-up were analyzed by one-way ANOVA followed by Tukey’s posttest. A *p* < 0.05 was considered to be statistically significant by using GraphPad Prism software.

## 3. Results

### 3.1. Molybdenum Concentration in Leaf Tissues 

The highest molybdenum leaf tissue concentration was detected in plants biofortified with a dosage of 3.0 µmol molybdenum L^−1^ (0.55 mg g^−1^ of dry weight) [23]. Thus, plants from plots treated with 3.0 µmol molybdenum L^−1^ were used for the nutritional intervention. The amount of molybdenum in control canasta lettuce was 0.21 mg/100 g fresh weight while the amount of molybdenum in biofortified canasta lettuce was 8 mg/100 g fresh weight.

### 3.2. Participants, Study Design and Compliance

The short pilot interventional study involved 24 volunteers aged 20–64 years (12 women, 12 men). All participants were in good general health and during the short period of molybdenum canasta lettuce administration. There was no evidence of differences between the groups (Table 2). All the participants ended the short time nutritional intervention after 12 days without any drop-out and best compliance. 

### 3.3. Lettuce Biofortified with Molybdenum and Glucose Metabolism

In the control group 12 days of lettuce consumption did not affect fasting glucose, insulin or insulin resistance compared with BASE. There were no differences in baseline glucose, insulin and insulin resistance and insulin sensitivity between the control (BASE) and the treated groups (BASE). In the treated group the consumption of lettuce biofortified with molybdenum significantly reduced fasting glucose, insulin, and insulin resistance with respect to BASE. The comparison between the groups (control vs treated) showed a significant reduction of fasting insulin, glucose and insulin resistance index (Figure 2A–C). Specifically, fasting glucose decreased from 83.4 ± 5.6 mg/dL (BASE) to 74.1 ± 4.3 mg/dL in the treated group at T2. Fasting insulin decreased from 9.1 ± 4.7 mUI/L (BASE) to 5.2 ± 1.8 mUI/L in the treated group at T2 while IR index decreased from 1.1 ± 0.6 (BASE) to 0.6 ± 0.2 in the treated group at T2. There was no difference in β-cell function in both the groups (control and treated) compared with BASE (Figure 2D). There was no difference in insulin sensitivity in the control group while in the treated group there was a significant increase in insulin sensitivity. The comparison between the groups (control vs treated) showed a significant increase in insulin sensitivity in the treated group compared to the control group at T2 (Figure 2E).

### 3.4. Gastrointestinal Hormones

In the control group there were no changes in endogenous levels of gastrointestinal hormones PYY, GIP, GLP-1, and GLP-2 after consumption for 12 days of canasta lettuce (Figure 3). In the treated group after 12 days of canasta lettuce PYY was significantly increased compared with BASE (Figure 3A). In addition, the GIP level was significantly increased after consumption of biofortified lettuce compared with BASE (Figure 3B). In both, the concentrations detected of the two gut peptides, PYY and GIP, were within the physiological range. Moreover, there was a significant change in endogenous levels of PYY and GIP in the treated compared to the control group (Figure 3A,B). The short nutritional intervention with molybdenum biofortified lettuce did not affect plasma GLP-1 and GLP-2 concentrations in the treated group compared to BASE. In addition, the comparison between the two groups (control vs. treated) showed no significant changes in GLP-1 and GLP-2 (Figure 3C,D).

### 3.5. Urinary Parameters

Data on molybdenum absorption ranged from 7.91 mg day^−1^ (99.0% of the intake via biofortified curly endive consumption) to 7.96 mg day^−1^ (99.6% of the molybdenum intake via biofortified lettuce consumption). Urinary samples did not show a significant increase in molybdenum concentration after 12 days of lettuce consumption (Table 3). 

## 4. Discussion

This is the first study that investigated the effects of a short nutritional intervention with molybdenum biofortified lettuce in individuals. The core purpose of this study was to investigate if consumption of lettuce biofortified with molybdenum would impact glucose homeostasis in a healthy population. In fact, we know that by following a healthy well-balanced diet it is possible to cover the intakes of micronutrients that meet the requirements for almost all the population, but we are aware that there are many situations in which the intake of macronutrients and micronutrients are less than adequate. This occurs in the healthy population as well as in overweight and obese people. There are values set for toxicity and deficiency of micronutrients but there may be additional benefits in the whole-body homeostasis if the intake is a little greater than needed to prevent deficiency. Therefore, it is of interest to verify the effect of biofortification with micronutrients in optimizing health, and in possible prevention of disease, similar to in this study. 

We did not find differences in dietary or lifestyle factors between the groups. In fact, in order to avoid bias in dietary intake (macronutrients, minerals, or vitamins) or lifestyle factors (e.g., physical activity) we did as follows: (1) we administered a questionnaire to collect anamnestic data of interest, food habits and lifestyle (2) we asked the cohort to follow the same regular nutritional pattern and lifestyle and do not use any food supplementation or integration 20 days before the baseline and for the entire period of 12 days of canasta lettuce administration (3) the cohort was provided with a food diary and instructed to record all food and beverages (including quantities) consumed over the 8 days before starting the study and until the end of the study. The results of this study suggest that consumption of biofortified vegetables with molybdenum could be a good strategy for preventing insulin resistance and diabetes in healthy individuals since consumption of lettuce biofortified with molybdenum ameliorates glucose homeostasis in healthy individuals with statistically significant difference. The results were confirmed by the lack of changes in glucose, insulin, insulin resistance or sensitivity in the control group that ate equal quantities of lettuce (not biofortified) for the same period. Specifically, the control group received 0.21 mg/100 g fresh weight of molybdenum a day for 12 days (control canasta lettuce) while the biofortified group received 8 mg/100 g fresh weight of molybdenum a day for 12 days (biofortified canasta lettuce). This value that could appear high if considered the tolerable UL for molybdenum (2 mg/day) but is fine. In fact, no negative effects were reported by the study population during the short-term nutritional intervention. Moreover, the biomonitoring equivalents (BEs) values for molybdenum associated with toxicity in urine range from 200 to 7500 μg/L while the molybdenum concentration we detected in urine ranged from 30 to 46 μg/L of molybdenum, in line to the BE values associated with exposure guidance values set to protect against both nutritional deficits and toxicity [16].

The consumption of 100 g a day of molybdenum biofortified lettuce, for a total of 12 days, was able to reduce fasting glucose, insulin levels and insulin resistance index in the treated group. These results are consistent, in literature, with a large case control study conducted in the Chinese population in which the higher plasma molybdenum was associated, in a dose-response manner, with reduced fasting plasma glucose and lower risk of metabolic syndrome [18]. Moreover, in pregnant women, during the late first trimester of pregnancy, lower molybdenum concentrations increased the risk of glucose dysregulation during pregnancy [17]. Therefore, it is possible to hypothesize that nutritional intervention with molybdenum biofortified lettuce could be potentially used on a population to maintain a stable glucose homeostasis in the body during different physiological states such as pregnancy or aging although further studies are required. 

The present study also showed that the consumption of molybdenum biofortified lettuce improved glucose metabolism by increasing insulin sensitivity but not by acting on β cell function. This was assessed by using the Homeostasis Model Assessment (HOMA) 2 model that allows estimating insulin sensitivity (% S) and β cell function (%β). From literature, clonal BRIN BD11 cells cultured with molybdate (1 mmol/L) showed enhanced responsiveness to glucose and enhanced basal insulin release [31]. Therefore, the consumption of molybdenum biofortified lettuce may improve insulin sensitivity by acting though a regulatory mechanism in which beta-cells are sensitized to secrete the minimum amount of insulin required to have an accurate glycemic control.

Following food intake, the gastrointestinal endocrine cells are selectively activated, on the basis of the different macronutrients and micronutrients composition of the meal, to secrete gut peptides [32] and in turn, these peptides influence glucose homeostasis [27,29,30,33,34,35]. Therefore, micronutrients present in the meal could act on endocrine cells by influencing the quantity of gut peptides released. In order to address this question gastrointestinal hormones released were measured in our plasma cohort of treated population finding no difference in GLP-1 and GLP-2 levels while PYY and GIP concentrations were significantly increased. The results suggest that PYY and GIP are involved in the mechanism of action by which molybdenum enriched lettuce acts to improve glucose homeostasis. This potential central role of PYY in this physiological mechanism of action is not surprising in consideration that PYY improves glycemic control [36] and suggests a possible PYY based therapy for the treatment of diabetes [37]. Regarding the role of GLP-1 and GIP it is known that the use of specific antagonists impairs glucose tolerance showing their additive effects in improving glucose tolerance [38]. At the beginning the idea was that both hormones required elevated concentrations of glucose to be effective [39] while studies on healthy individuals, showed that both peptides acted, about equally, on insulin secretion already at fasting glucose levels [40]. However, there is an important difference between GLP-1 and GIP, because GLP-1 inhibits glucagon secretion [41] while GIP seems influencing, in quantitative way, insulin signaling [39]. This could explain because, in our experiments, GLP-1 levels did not change while GIP levels increased. However, further studies in the field are required due to the complexity of the system. In fact, in type 2 diabetes GLP-1 maintains its stimulatory activity while GIP loses it [39] and this is still unexplained.

With respect to urinary molybdenum concentration, we did not see any difference in the treated population, although we detected a wide fluctuation of measurements within both cohorts (control and treated). There are several explanations that we can count on. The whole amount of molybdenum was absorbed since adults absorb from 40% to 100% of dietary molybdenum [42]. The amount of molybdenum was administered for a too short period. The molybdenum was removed very rapidly from the circulation. In fact, urinary excretion of molybdenum is higher and quicker in the few hours after administration and after 24 h molybdenum excretion rates are insignificant [43].

We are conscious that there are some limitations in the study including the short duration of the intervention and the small sample size. However, a key strength of the intervention protocol was to assess the effects of the intervention within and between the two groups.

## 5. Conclusions

The study showed that a short nutritional intervention with molybdenum biofortified lettuce on a population of healthy individuals ameliorates glucose homeostasis by influencing gastrointestinal PYY and GIP levels. These results confirm how biofortified food may be used as a tool in disease prevention.

## Figures and Tables

**Figure 1 nutrients-14-01351-f001:**
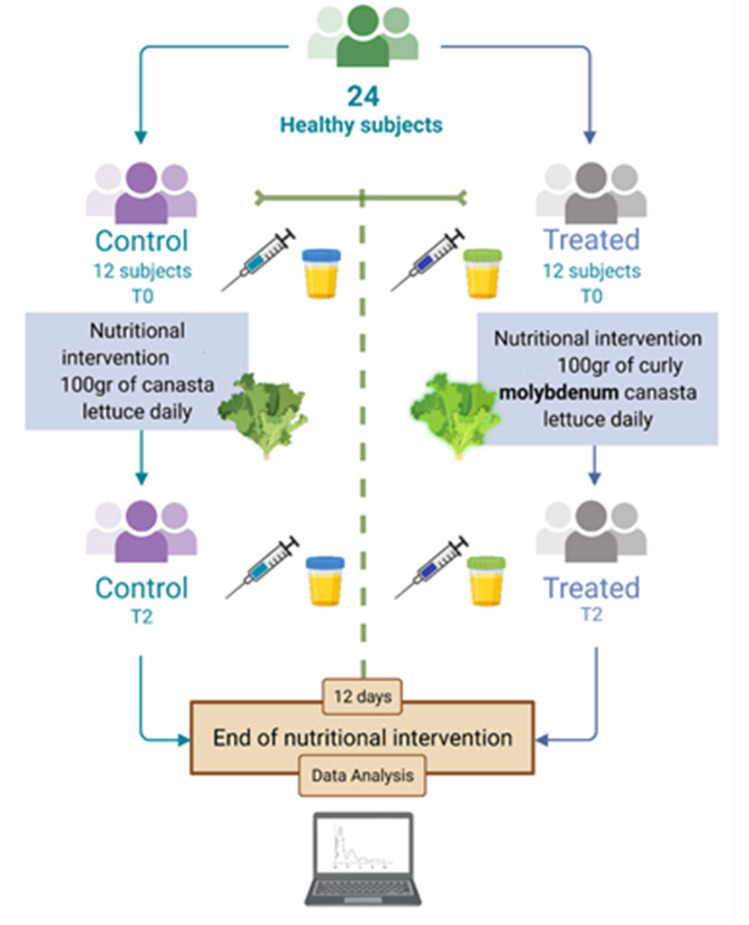
Flow chart of the interventional nutritional study during the period of 12 days of administration of biofortified molybdenum canasta lettuce and lettuce without any biofortification. Control = control group eating canasta lettuce with no biofortification, Treated = treated group eating lettuce biofortified with molybdenum.

**Figure 2 nutrients-14-01351-f002:**
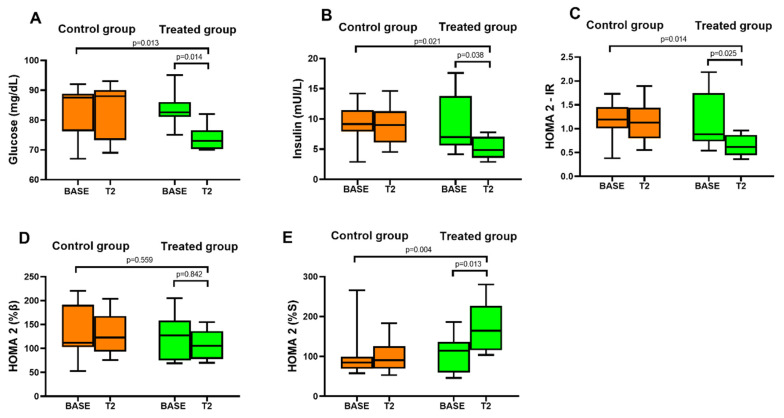
Markers of glucose metabolism measured at baseline (BASE) and after 12 days of lettuce consumption in the control group and in the treated group. (**A**) Box and whisker plot of fasting glucose. (**B**) Box and whisker plot of fasting insulin. (**C**) Box and whisker plot of insulin resistance. (**D**) Box and whisker plot of β-cell function. (**E**) Box and whisker plot of insulin sensitivity.

**Figure 3 nutrients-14-01351-f003:**
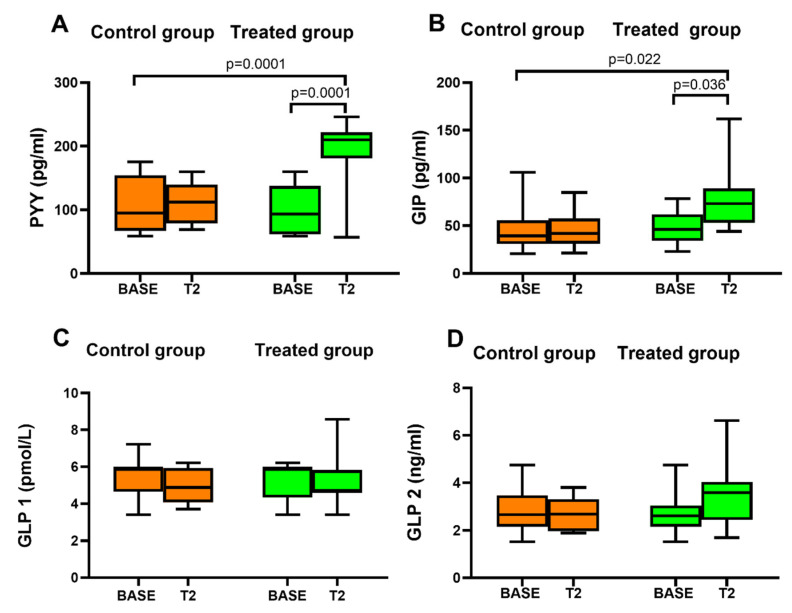
Gastrointestinal peptide concentrations measured at baseline (BASE) and after 12 days of lettuce consumption in the control group and in the treated group (**A**) Box and whisker plot of Peptide YY (PYY). (**B**) Box and whisker plot of Gastric Inhibitory Polypeptide (GIP). (**C**) Box and whisker plot of Glucagon-Like Peptide 1 (GLP-1). (**D**) Box and whisker plot of Glucagon-Like Peptide 2 (GLP-2).

**Table 1 nutrients-14-01351-t001:** Inclusion and exclusion criteria of the study.

Inclusion Criteria	Exclusion Criteria
Italian ethnicity	Chronic disease
Age: 18–60 years	Intake of drugs, vitamins or dietary supplements
Clinically healthy	Pregnancy, breastfeeding, exogenous hormones
Body mass index between 18.5 and 28 kg/m^2^	

**Table 2 nutrients-14-01351-t002:** Characteristics of the subjects in the two groups of study. W = weight, H = height, BMI = body mass index, FM = fat mass, MM = muscle mass, VF = visceral fat; ND = not statistically different from each other. Values are means ± SD. *n* = 12 in each group.

Characteristics of the Subjects	Control Group	Treated Group	
	Mean ± SD	Mean ± SD	*p*-Value
W (Kg)	69 ± 9.8	72 ± 13	ND
H (cm)	168 ± 10	172 ± 9	ND
BMI	24.3 ± 2.5	24.2 ± 2.8	ND
FM (%)	28.6 ± 7.3	26.5 ± 7.1	ND
MM (%)	32.8 ± 5.8	33.2 ± 5.4	ND
VF (%)	6.5 ± 3.3	7.2 ± 3.2	ND

**Table 3 nutrients-14-01351-t003:** Urinary molybdenum levels measured at baseline (BASE) and after 12 days of lettuce consumption (T2) in the control and in the treated group. ND = not statistically different from each other. Values are means ± SD. *n* = 12 in each group.

Urinary Molybdenum Concentration (μg/L)						
Control Group						
BASE			T2			
Mean	SD	*n*	Mean	SD	*n*	*p* Value
37.02	25.62	12	45.946	29.443	12	ND
Treated Group						
BASE			T2			
Mean	SD	*n*	Mean	SD	*n*	*p* Value
42.253	28.035	12	31.831	21.789	12	ND

## Data Availability

The datasets during and/or analyzed during the current study are available from the corresponding authors on reasonable request.

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
