# Peer review of "Impact on Glucose Homeostasis: Is Food Biofortified with Molybdenum a Workable Solution? A Two-Arm Study"

_nutrients, 2022, doi:10.3390/nu14071351_

Round 1
Reviewer 1 Report
Reviewer comments and suggestions
The present research paper has uncovered the impact of molybdenum on glucose homeostasis. The study suggests that the consumption of lettuce biofortified with molybdenum ameliorates glucose homeostasis in healthy individuals and also showed that the consumption of molybdenum biofortified lettuce for twelve days reduced fasting glucose, insulin, glucose, insulin resistance, and increased insulin sensitivity but did not affect beta cell function. Finally the authors suggested that consumption of lettuce biofortified with molybdenum improves glucose homeostasis and PYY and GIP are involved in the mechanism of action.
The paper has nicely complied and covered the utmost part related to Molybdenum in glucose metabolism. However, in many places, the authors need to do a few small corrections in the manuscript. Based on my view, below are the comments that need to be incorporated in the revised version of the manuscript.
- Line no 56; [Baldassano et al, 2022] cited in proper way.
- Write the full form of various abbreviations used more than one time in MS (Ex. PYY, GLP-1, GIP, and GLP-2). Both in abstract and text needed the full form.
- Line 57; removed comma from sentence 57-58 (The source of many important minerals is, mainly, plant‐based, therefore it is important to use a balanced diet to provide an adequate proportion of minerals).
- Line 80 (recommended Dietary Allowance), use small letter for Dietary Allowance in sentence.
- Line 95; (2.1. Trial setup, plant materials, Nutraceutical Traits, and crop management) use small latter for Nutraceutical Traits.
- Use the same pattern for writing figure in the text ((figure 1 or Fig. 1)
- Figure 3 legend needed full form of the parameter used in the analysis.
- Table 3, T2 stand for, please write in the legend part
- Introduce the abbreviation of Molybdenum in abstract instead of line 59 and then use it everywhere. (Mo)
- How this study is can be implemented? What is the core purpose of this study? As you have mentioned in lines 69-72, Mo deficiency is not very common.
- The first para of discussion does not acquire novel information, better to delete 4-5 lines
- Line 317, metabolic syndrome not Metabolic syndrome, how did the authors check “not by acting on β cell function”.
- Reference 37 need to be explored. Line 344-346 needs to be rewritten.
- The journal style of few references are not correct. Such as 7,9,10,14,23,25,33, and 36
Author Response
We would like to thank the referee very much for the comments to further improve the manuscript.
- Line no 56; [Baldassano et al, 2022] cited in proper way.
Done. Sorry. Please refer to line 57, [9]
- Write the full form of various abbreviations used more than one time in MS (Ex. PYY, GLP-1, GIP, and GLP-2). Both in abstract and text needed the full form.
Done. Please refer to lines 30-31 and 94-95.
- Line 57; removed comma from sentence 57-58 (The source of many important minerals is, mainly, plant‐based, therefore it is important to use a balanced diet to provide an adequate proportion of minerals).
Done. Please refer to lines 57-59
- Line 80 (recommended Dietary Allowance), use small letter for Dietary Allowance in sentence.
Done. Please refer to line 81.
- Line 95; (2.1. Trial setup, plant materials, Nutraceutical Traits, and crop management) use small latter for Nutraceutical Traits.
Done. Please refer to line 97.
- Use the same pattern for writing figure in the text ((figure 1 or Fig. 1)
Done.
- Figure 3 legend needed full form of the parameter used in the analysis.
We added the full form of the parameters as you suggested in figure 3.
- Table 3, T2 stand for, please write in the legend part
Done, “Urinary molybdenum levels measured at baseline (BASE) and after 12 days of lettuce consumption (T2) in the control and in the treated group. Values are means ± SD. n = 12 in each group”
- Introduce the abbreviation of Molybdenum in abstract instead of line 59 and then use it everywhere. (Mo)
Thanks. We decided to use Molybdenum instead of the abbreviation (Mo) and we used it everywhere.
- How this study is can be implemented? What is the core purpose of this study? As you have mentioned in lines 69-72, Mo deficiency is not very common.
Thank you very much for the point. In this study we tested if consumption of lettuce biofortified with molybdenum was able to ameliorate glucose homeostasis in the healthy population. In fact, we know that by following a healthy well-balanced diet it is possible to cover the intakes of micronutrients that meet the requirements for almost all the population but we are aware that there are many situations in which the intake of macronutrients and micronutrients are less than adequate. This occurs in the healthy population as well as in overweight and obese people. There are values set for toxicity and deficiency of micronutrients but there may be additional benefits in the whole-body homeostasis if the intake is a little greater than needed to prevent deficiency. Therefore, it is of interest to verify the effect of biofortification with micronutrients in optimizing health, and in prevention of disease like in this study. As you suggested we have added the point in lines 282-292.
- The first para of discussion does not acquire novel information, better to delete 4-5 line.
As you suggested we have deleted the first lines of the discussion.
- Line 317, metabolic syndrome not Metabolic syndrome, how did the authors check “not by acting on β cell function”.
Done. We checked it by measuring (%β) by using the Homeostasis Model Assessment (HOMA) 2 model that allow us to estimate β cell function. Thanks for the point. We added it in the discussion lines 324-325.
- Reference 37 need to be explored. Line 344-346 needs to be rewritten.
As you suggested the reference has been explored and the sentence has been rewritten.
Please refer to lines 342-353.
- the journal style of few references are not correct. Such as 7,9,10,14,23,25,33, and 36
Thank you. We have corrected the style of the references.
Reviewer 2 Report
The article of Sara Baldassano and co-workers outlines the impact of consumption of biofortified food with molybdenum on glucose homeostasis. This is an interesting article with adequate information that deserves to be published. After resolving the following criteria, it may be accepted for publishing.
Page 2 line 56, a reference must be added with the title of the article, the authors and the journal in which it was or will be submitted in 2022 but cannot be indicated in this way.
Page 2 the meaning of the acronym (EAR) must be given as soon as it appears on line 75 and not on line 78.Capital letters line 80 Dietary Allowance don't seem necessary to me
Page 4 the table should be placed after line 139 after the sentence "We advise the cohort to follow.....pattern" at the end of this paragraph for better understanding.Capitals are not necessary line 171 for Barefoot.. Body mass index.. Body composition.Page 5, the chemical compounds must be written with the number in subscript HNO3 and not HNO3 or H2O2 and not H2O2 line 186, line 194 and line 202.The content of paragraphs 2.3 and 2.4 must be justified alignment.Page 6, line 245 (Table.2) dot must be removed.Page 7, Figure 2 must be bold.In references part, doi are missing for some references 3,5,10,16,17,20,21,23,26,29 and must be added.
Author Response
Thank you very much for your comments and suggestions.
Page 2 line 56, a reference must be added with the title of the article, the authors and the journal in which it was or will be submitted in 2022 but cannot be indicated in this way.
Done. Sorry. Please refer to line 57, [9].
Page 2 the meaning of the acronym (EAR) must be given as soon as it appears on line 75 and not on line 78.Capital letters line 80 Dietary Allowance don't seem necessary to me
Done. Please refer to line 77 and 81
Page 4 the table should be placed after line 139 after the sentence "We advise the cohort to follow.....pattern" at the end of this paragraph for better understanding.
Done. Please refer to line 139.
Capitals are not necessary line 171 for Barefoot.. Body mass index.. Body composition.Page 5, the chemical compounds must be written with the number in subscript HNO3 and not HNO3 or H2O2 and not H2O2 line 186, line 194 and line 202.
Done. Please refer to lines 168-170 and 174, 181, 189.
The content of paragraphs 2.3 and 2.4 must be justified alignment.
Done.
Page 6, line 245 (Table.2) dot must be removed.
Done.
Page 7, Figure 2 must be bold.
Done.
In references part, doi are missing for some references 3,5,10,16,17,20,21,23,26,29 and must be added.
Thanks. We added doi.
Reviewer 3 Report
Summary of manuscript: This study aimed to assess whether the consumption of lettuce fortified with molybdenum influences glucose homeostasis and selected gastrointestinal hormone levels. Twenty-four individuals were supplemented with molybdenum-fortified lettuce for 12 days. The results demonstrated that the consumption of lettuce fortified with molybdenum significantly decreased fasting glucose, insulin, insulin resistance, and increased insulin sensitivity, but did not impact beta cell function in healthy individuals. Urinary concentrations of molybdenum were not significantly different. It was concluded that lettuce fortified with molybdenum improves glucose homeostasis, with PIY and GIP involved in the mechanisms of action.
General comments: I carefully reviewed this manuscript. The authors provided a study with interesting results. However, there are many spelling and grammatical errors that make the manuscript very difficult to read. I recommend including more details in the manuscript. I provided my specific comments below.
Abstract
Point 1: Lines 30-31: I suggest stating “Blood and urine samples”
Point 2: Line 33: I suggest stating “Urine samples”
Point 3: Line 36: Should “glucose” be stated again? It was indicated on line 35.
Point 4: Lines 37-38: This sentence is difficult to read, as “concentration” was stated twice. Please revise.
Introduction
Point 5: Line 44: “T2D” should be in parentheses.
Point 6: Line 45-46: “changing in lifestyle” needs to be revised.
Point 7: Line 50: “that could hold the cost of healthcare” needs to be revised. I don’t understand this sentence.
Point 8: Line 56: This citation is not properly formatted.
Point 9: Line 64: What is the definition of Moco?
Point 10: Lines 64-66: Please identify which enzymes are responsible for these metabolic reactions.
Point 11: Line 67: Legume should be plural.
Point 12: Line 70: This should be Crohn’s disease.
Point 13: Line 71: Should this be irritable bowel syndrome?
Point 14: Line 74: This line is difficult to understand. Please revise.
Point 15: Line 77: molybdenum should not be capitalized. Please check throughout manuscript.
Point 16: Line 80: Recommended should be capitalized with (RDA).
Point 17: Line 81: “the” should be removed.
Point 18: Line 92: “changes” should be used.
Point 19: Line 93: Please define PYY, GLP-1, GIP, and GLP-2.
Materials and Methods
Point 20: Line 95: Nutraceutical Traits should not be capitalized.
Point 21: Line 96: Is “iodine” correct?
Point 22: Line 102: “leaves tissues” needs to be revised.
Point 23: Figure 1: What is “whit”?
Point 24: Line 139: “advised” should be included.
Point 25: Lines 149-150: “male” should be plural. Were the subjects randomly assigned to the groups?
Point 26: Line 171: Barefoot and Body should not be capitalized.
Point 27: Line 189: Should “In” be capitalized?
Point 28: Lines 193-194: molybdenum is not spelled correctly. Please check throughout manuscript.
Point 29: Line 196: Please define SD.
Point 30: Line 217: “previusly” should be changed to “previously”
Point 31: Why was the treatment duration for 12 days? Please indicate in the Methods section.
Results
Point 32: Line 244: Please be consistent using “Mo” throughout the manuscript.
Point 33: Lines 248-249: Change to “weight”. “MS” is used in the Table 2 legend, but not in Table 2. BMI is not included in the legend. Please revise.
Point 34: Figure 2: Were there any differences in baseline glucose, insulin, insulin resistance, etc. between the control and treated groups?
Point 35: Table 3: Please include statistical information in the table and include a legend. Based on Table 3, Mo urinary concentrations increased in the control group, and decreased in the treated group. Is this correct?
Discussion
Point 36: Line 303: Include “of a short”
Point 37: Line 307: Remove “on”
Point 38: Line 312: Specifically, how much molybdenum was consumed by each group? This should be stated in the Results and Discussion sections.
Point 39: Line 314: Please revise “heath treated population”
Point 40: Were there any differences in other nutrient intakes (vitamins, minerals, and macronutrients) between the control and treatment groups? This should be covered in the Discussion.
Point 41: How do you know that the results are from molybdenum and not from other differences between the groups (dietary, exercise, and/or other lifestyle factors)?
Author Response
We would like to thank the referee for the valuable suggestions to improve the clarity of the manuscript.
Abstract
Point 1: Lines 30-31: I suggest stating “Blood and urine samples”
Done. Please refer to line 33.
Point 2: Line 33: I suggest stating “Urine samples”
done. Please refer to line 35.
Point 3: Line 36: Should “glucose” be stated again? It was indicated on line 35.
Done.
Point 4: Lines 37-38: This sentence is difficult to read, as “concentration” was stated twice. Please revise.
Done. The sentence was revised as you suggested. Please refer to line 39-40
Introduction
Point 5: Line 44: “T2D” should be in parentheses.
Done. Please refer to line 47.
Point 6: Line 45-46: “changing in lifestyle” needs to be revised.
The sentence was revised as you suggested. Please refer to line 47-49.
Point 7: Line 50: “that could hold the cost of healthcare” needs to be revised. I don’t understand this sentence.
The sentence was revised as you suggested. Please refer to lines 51-52.
Point 8: Line 56: This citation is not properly formatted.
We have corrected it. Please refer to line 57, citation [9].
Point 9: Line 64: What is the definition of Moco?
The definition of Moco is molybdenum cofactor (Moco). The sentence was revised according to your suggestion. Please refer to line 66
Point 10: Lines 64-66: Please identify which enzymes are responsible for these metabolic reactions.
Done. Please refer to lines 64-68.
Point 11: Line 67: Legume should be plural.
Done. Please refer to line 70.
Point 12: Line 70: This should be Crohn’s disease.
Done. Please refer to line 74.
Point 13: Line 71: Should this be irritable bowel syndrome?
The sentence was unclear and has been revised. Please refer to lines 73.
Points 14: Line 74: This line is difficult to understand. Please revise.
The sentence was revised as you suggested. Please refer to lines 74-76
Point 15: Line 77: molybdenum should not be capitalized. Please check throughout manuscript.
We have checked throughout the manuscript and corrected it.
Point 16: Line 80: Recommended should be capitalized with (RDA).
Done. Please refer to line 81.
Point 17: Line 81: “the” should be removed.
Done. Please refer to line 82.
Point 18: Line 92: “changes” should be used.
Done. Please refer to line 93.
Point 19: Line 93: Please define PYY, GLP-1, GIP, and GLP-2.
Done. Please refer to lines 94-95.
Materials and Methods
Point 20: Line 95: Nutraceutical Traits should not be capitalized.
Done. Please refer to line 97.
Point 21: Line 96: Is “iodine” correct?
No is molybdenum. We have corrected it. Please refer to line 98. Thank you very much.
Point 22: Line 102: “leaves tissues” needs to be revised.
Done: “leaves”. Please refer to line 104.
Point 23: Figure 1: What is “whit”?
Done: Please refer to line 128.
Point 24: Line 139: “advised” should be included.
Done. Please refer to line 137
Point 25: Lines 149-150: “male” should be plural. Were the subjects randomly assigned to the groups?
Yes, the subjects were randomly assigned to the groups. The sentence was added to clarify it. Please refer to lines 150- 152
Point 26: Line 171: Barefoot and Body should not be capitalized.
Done. Please refer to line 168-169.
Point 27: Line 189: Should “In” be capitalized?
No. Done. Please refer to line 176. Thanks
Point 28: Lines 193-194: molybdenum is not spelled correctly. Please check throughout manuscript.
Done. Thanks.
Point 29: Line 196: Please define SD.
Done. Please refer to line 184.
Point 30: Line 217: “previusly” should be changed to “previously”
Done. Please refer to line 198.
Point 31: Why was the treatment duration for 12 days? Please indicate in the Methods section.
The treatment duration was for 12 days because our previous study showed no negative effect after 12 days of nutritional intervention with Iodine biofortified curly endive. As you suggested this was indicated in the methods section, please refer to lines 148-150.
Results
Point 32: Line 244: Please be consistent using “Mo” throughout the manuscript.
Thanks. We decided to use molybdenum instead of “Mo” throughout the manuscript. We have double checked it but if we missed something please let us know and we will modify it immediately.
Point 33: Lines 248-249: Change to “weight”. “MS” is used in the Table 2 legend, but not in Table 2. BMI is not included in the legend. Please revise.
Done. Please refer to line 224-225.
Point 34: Figure 2: Were there any differences in baseline glucose, insulin, insulin resistance, etc. between the control and treated groups?
No, there were no differences in baseline glucose, insulin and insulin resistance and insulin sensitivity between the control (BASE) and the treated groups (BASE).
|
Parameters |
Control group (BASE) mean and SD |
Tretated group (BASE) mean and SD |
p value |
|
Glucose (mg/dL) |
83.5 ± 8 |
83.4 ± 5.6 |
ND |
|
Insulin (mU/L) |
9.4 ± 3 |
9.1 ± 4.7 |
ND |
|
IR-index |
1.2 ± 0.4 |
1.1 ± 0.6 |
ND |
|
% β |
132 ± 53 |
124 ± 47 |
ND |
|
% S |
98 ± 55 |
106 ± 45 |
ND |
Point 35: Table 3: Please include statistical information in the table and include a legend. Based on Table 3, Mo urinary concentrations increased in the control group, and decreased in the treated group. Is this correct?
There was a tendency of increasing but did not reach any statistical significance. As you suggested we have included statistical information and legend in table 3.
Discussion
Point 36: Line 303: Include “of a short”
Done. Please refer to line 281.
Point 37: Line 307: Remove “on”
Done. Please refer to lines 283.
Point 38: Line 312: Specifically, how much molybdenum was consumed by each group? This should be stated in the Results and Discussion sections.
The control group received 0.21 mg/100 g fresh weight of molybdenum a day for 12 days by consuming the control canasta lettuce while the biofortified group received 8 mg/100 g fresh weight of molybdenum a day for 12 days by consuming the biofortified canasta lettuce. As you required the information was added in the results and discussion section. Please refer to lines 213-215 and to lines 308-311.
Point 39: Line 314: Please revise “heath treated population”
Done. Please refer to line 313.
Point 40 and Point 41: Were there any differences in other nutrient intakes (vitamins, minerals, and macronutrients) between the control and treatment groups? This should be covered in the Discussion. How do you know that the results are from molybdenum and not from other differences between the groups (dietary, exercise, and/or other lifestyle factors)?
Thank you very much for these two interesting questions. In order to avoid that the results were from differences in dietary (nutrient intake of macronutrients, minerals, or vitamins) or lifestyle factors (e.g physical activity) between the groups 1) we administered a questionnaire to collect anamnestic data of interest, food habits and life-style 2) We asked the cohort to follow the same regular nutritional pattern and life style and do not use any food supplementation or integration 20 days before the baseline and for the entire period of 12 days of canasta lettuce administration. 3) The cohort was provided with a food diary and instructed to record all food and beverages (including quantities) consumed over the 8 days before starting the study and until the end of the study. This first period was assumed to be representative of the healthy volunteer’s habitual nutritional intake. There were no differences in dietary of life-style factors between the groups. The two groups were homogeneous as reported also by anthropometric measurements. Moreover, the presence of differences within the treated group (same people before and after consumption of biofortified lettuce) and the lack of differences within the control group (same people before and after consumption of lettuce) indicate that the effects within and between the groups were from molybdenum. As you suggested we have added the point to the discussion. Please refer to lines 293-301.
Round 2
Reviewer 3 Report
I would like to thank the authors for addressing many of my concerns. However, I suggest additional changes. I provided my specific comments below.
Point 1: Title, Line 2: I suggest changing “molybdenum food” to food biofortified with molybdenum or a similar change.
Abstract
Point 2: Line 27: Please change to “would be due”
Point 3: Line 37: Please change to “in healthy people”
Introduction
Point 4: Line 49: Please change to “approach”
Point 5: Line 49: I suggest changing this sentence to “that could help in reducing drug prescriptions and hospitalizations”
Point 6: Line 61: Should this be “mammals”?
Point 7: Line 62: Should this be “sulfite”?
Point 8: Line 75: Institute of Medicine should be capitalized.
Point 9: Line 81: The tolerable upper intake level (UL) for molybdenum is 2 mg/day. This should be mentioned.
Materials and Methods.
Point 10: Figure 1: I don’t understand what the word “whit” means in Figure 1. The word “whit” appears in the two light blue boxes.
Point 11: Line 129: The age range is 20-64; however, Table 1 indicates 18-60 years. Please clarify.
Point 12: Table 1: Breastfeeding should not be capitalized in Table 1.
Point 13: Line 142: Declaration should be capitalized. Please change to “It was approved”
Point 14: Line 146: Did the participants return the lettuce plants that were not eaten?
Point 15: Line 148: Iodine should not be capitalized.
Point 16: Line 164: Please change to “urine samples”
Point 17: Line 167: Please change “was” to were.
Point 18: Line 188: “determined” should not be bolded.
Point 19: Line 191: Please change “previusly” to previously.
Point 20: Line 196: Please change to “measured”
Results
Point 21: Table 2 legend, Line 224: Please include the definition for ND.
Point 22: Line 235: Please change “form” to from.
Point 23: Line 241: Based on the previous response document, please include your following statement: “There were no differences in baseline glucose, insulin and insulin resistance and insulin sensitivity between the control (BASE) and the treated groups (BASE).”
Point 24: Line 251: Compared to what?
Point 25: Figure 3A: Should there be a statistical bracket for p=0.0001 regarding BASE compared to T2 for the treated group?
Point 26: Figure 3B: Please change “Treated group group” to Treated group.
Point 27: Lines 269-271: Is this sentence correct? Should the control diet be included here regarding molybdenum intake?
Point 28: Table 3 legend, Line 275: Please include the definition for ND.
Point 29: Table 3: To confirm: there is no statistical significant difference between 42.253 and 31.831 regarding urinary molybdenum concentrations for the treated group?
Discussion
Point 30: Line 293: Please change to “as follows”
Point 31: Line 295: Please change to “lifestyle”
Point 32: Line 300: “results of this study” should not be bolded.
Point 33: Line 302: “since” should not be bolded.
Point 34: Line 308: It was indicated that the participants consumed 8 mg/day of molybdenum. The Tolerable Upper Intake Level (UL) for molybdenum is 2 mg/day. Therefore, the treated group consumed 4 times the UL, which increases the risk for toxicity. Was this a concern for the Ethic Committee of Palermo University Hospital? These topics should be discussed around line 309.
Point 35: Line 334: Please change to “released”
Point 36: Line 346: Please change to “inhibits”
Conclusion
Point 37: Line 365: Please change to “a short”
Author Response
Thank you very much for your comments and suggestions.
Point 1: Title, Line 2: I suggest changing “molybdenum food” to food biofortified with molybdenum or a similar change.
As you suggested the title has been chanced to “food biofortified with molybdenum” Please refer to line 2.
Abstract
Point 2: Line 27: Please change to “would be due”
Done. Please refer to line 29
Point 3: Line 37: Please change to “in healthy people”
Done. Please refer to line 39
Introduction
Point 4: Line 49: Please change to “approach”
Done. Please refer to line 50
Point 5: Line 49: I suggest changing this sentence to “that could help in reducing drug prescriptions and hospitalizations”
Done. Please refer to line 51
Point 6: Line 61: Should this be “mammals”?
Done. Please refer to line 62
Point 7: Line 62: Should this be “sulfite”?
Done. Please refer to line 64
Point 8: Line 75: Institute of Medicine should be capitalized.
Done. Please refer to line 77
Point 9: Line 81: The tolerable upper intake level (UL) for molybdenum is 2 mg/day. This should be mentioned.
Done. Please refer to line 83-84
Materials and Methods.
Point 10: Figure 1: I don’t understand what the word “whit” means in Figure 1. The word “whit” appears in the two light blue boxes.
The word “whit” has been deleted from the figure 1. Thanks.
Point 11: Line 129: The age range is 20-64; however, Table 1 indicates 18-60 years. Please clarify.
20-64 was the age range of our cohort of study while 18-60 indicated the ages to be included in the study. We deleted 20-64 from line 125 because the information is not necessary. Thanks.
Point 12: Table 1: Breastfeeding should not be capitalized in Table 1.
done
Point 13: Line 142: Declaration should be capitalized. Please change to “It was approved”
Done. Please refer to line 137-138.
Point 14: Line 146: Did the participants return the lettuce plants that were not eaten?
no
Point 15: Line 148: Iodine should not be capitalized.
done
Point 16: Line 164: Please change to “urine samples”
Done. Please refer to line 160.
Point 17: Line 167: Please change “was” to were.
Done. Please refer to line 168
Point 18: Line 188: “determined” should not be bolded.
done
Point 19: Line 191: Please change “previusly” to previously.
Done. Please refer to line 187
Point 20: Line 196: Please change to “measured”
Done. Please refer to line 191
Results
Point 21: Table 2 legend, Line 224: Please include the definition for ND.
Done. Please refer to line 219-220
Point 22: Line 235: Please change “form” to from.
Done. Please refer to line 231
Point 23: Line 241: Based on the previous response document, please include your following statement: “There were no differences in baseline glucose, insulin and insulin resistance and insulin sensitivity between the control (BASE) and the treated groups (BASE).”
Done. Please refer to line 225-226
Point 24: Line 251: Compared to what?
The sentence has been corrected. Thanks.
Point 25: Figure 3A: Should there be a statistical bracket for p=0.0001 regarding BASE compared to T2 for the treated group?
The figure 3 has been adjusted. Thanks.
Point 26: Figure 3B: Please change “Treated group group” to Treated group.
The figure B3 has been changed. Thanks.
Point 27: Lines 269-271: Is this sentence correct? Should the control diet be included here regarding molybdenum intake?
Done. Please refer to line 266-268
Point 28: Table 3 legend, Line 275: Please include the definition for ND.
Done. Please refer to line 272
Point 29: Table 3: To confirm: there is no statistical significant difference between 42.253 and 31.831 regarding urinary molybdenum concentrations for the treated group?
Yes, not statistically different from each other.
Discussion
Point 30: Line 293: Please change to “as follows”
Done. Please refer to line 291
Point 31: Line 295: Please change to “lifestyle”
done
Point 32: Line 300: “results of this study” should not be bolded.
done
Point 33: Line 302: “since” should not be bolded.
done
Point 34: Line 308: It was indicated that the participants consumed 8 mg/day of molybdenum. The Tolerable Upper Intake Level (UL) for molybdenum is 2 mg/day. Therefore, the treated group consumed 4 times the UL, which increases the risk for toxicity. Was this a concern for the Ethic Committee of Palermo University Hospital? These topics should be discussed around line 309.
Thank you very much for the point. As you state this value that could appear high if considered the tolerable UL for molybdenum (2 mg/day) is fine. In fact, no negative effects were reported by the study population during the short-term nutritional intervention. Moreover, the biomonitoring equivalents (BEs) values for molybdenum associated with toxicity in urine range from 200 to 7500 μg/L while the molybdenum concentration we detected in urine ranged from 30 to 46 μg/L of molybdenum, in line to the BE values associated with exposure guidance values set to protect against both nutritional deficits and toxicity. As you suggested this was added to the discussion lines 306-312.
Point 35: Line 334: Please change to “released”
Done. Please refer to line 339.
Point 36: Line 346: Please change to “inhibits”
Done. Please refer to line 351
Conclusion
Point 37: Line 365: Please change to “a short”
done